# ML-Approach to Qualimetry: Building Property Trees and Calculating Value Weights using Generative Neural Networks

## Abstract

For sustainable Integrated Territorial Development (ITD), evaluating multifaceted value (social, ecological, etc.) is necessary, beyond traditional economic cost. Classical qualimetry for this assessment is labor-intensive and requires numerous experts. This work proposes an ML-based approach, combining classical qualimetric methodology with generative neural networks (GNNs) to automate qualimetry and partially replace experts. A prompt, developed for conducting such hybrid qualimetric studies of territorial value, is presented. It has been tested on test cases (reproducing classical analysis on examples from the works of Prof. Azgaldov G.G.). Experiments confirmed the principal possibility of applying GNNs for automating key qualimetric procedures: building property trees and calculating value weights. Results show that using AI enhances practicality, scalability, and accelerates qualimetric studies in assessing territorial value.

Projects of the integrated development of the built -up territories (CCT) are a powerful comprehension of the transformation of the urban space, aimed at the creature -composed, stable and economically prosperous environment. Traditionally, they are mainly on the economic component - the cost of icapons. Such an approach focused on measurable financial indicators can lead to a missing from other, no less important aspects that determine the urban environment and the well -being of its inhabitants.

The monograph [6] proposes an approach to the implementation of KRT projects, which is based on the concept of rational and most effective use (RNEI) [5]. This concept is characteristic of the theory and practice of land management, but it is not used very rarely used by developers and urban planners. The concept is based on flooring the value of the territory, and not its value. In contrast to the value of value, the value of the territory should take into account such important aspects as social, environmental, cultural, aesthetic factors, etc. The underestimation of these factors can cite to krojects that are effective from an economic point of view, but have negative -headed consequences for the urban environment and its inhabitants, which is critical for sustainability.

The problem of assessing the value of real estate was fundamentally investigated by the Sarah Seis gateway and its colleagues from the school of the cereweigers of the Kingstonnaya University "Assessment of real estate: from value to value" [7].

The combination of various values of stakeholders into a single integral indicator is a difficult task [6], which requires the accounting of various points and the use of methods such as multicriterial analysis and determination coefficients. . Different groups of stakeholders (for example, developers and local livers) can have conflicting ideas about value. The analysis conducted in [6] tried that it is

not possible to make a correct assessment of the value of the territory with instruments of economic measurements [3]. For this procedure, it is necessary to apply the methods of quantitative quality assessment, in particular the method of text.

Qualimetry, scientific discipline engaged in quantitative assessment of the formation [1, 2, 8], was formed back in the 60s of the last century, its active conductor in the practice of measurements was prof. Azgaldov G.G. However, the theory of qualimetry did not receive wide expansion. First of all, due to the complexity of research.

It is laborious and requires the involvement of many experts, often with a separation from Ikhosnovaya activity for a considerable time. To develop a methodology for evaluating the formation (IOC), the creation of an organizational, technical and expert groups is required. The estimate of the expert group can reach 7-10 people, sometimes creative-haired such groups are required. The above algorithm turned out to be very expensive and laborious, since it was formed without taking into account modern technologies.

The emergence of artificial intelligence (AI), especially generative neural networks (STS), opening a damage to the labor of labor capacityquantitative quality assessment [4]. STS, such as DeepSeek and Gemini 2.0 Flash, demonstrated a high level of training and expertia. STS is able to incite and generate text and code similar to human, as well as execute.

Tasks related to the reasoning and processing of information. They are possessions, relevant qualimetry analysis, including understanding of the text, extraction of information and recognition of images. AI is already used in various analytical tasks and expert areas, such as medical diagnostics or analysis of legal documents. The main problem of the researcher when working with the correct task for the work of the STS, which is called Promt. The development of effective interactive Promov is a strategy for the STS department in the implementation of qualimetric analysis at the expert level. Properly developed prosts provide context, instructions and restrictions on the model. Despite their capabilities, modern STS have restrictions. They can admit inaccuracies ("hallucinations"), inherit displacements from training data, experience difficulties with complex numerical reasoning and do not have true beings or common sense. The translation of multifaceted judgments and implicit knowledge of experts into explicit rules for AI is a significant problem and for a time cannot be fully automated and require the verification of the transmission data of the model.

To work out the qualimetric Promt among the Monica STS integrator with the sincoling of the DEPSEEK V3 STS, a bot named "Kalimetry" was created. Work on qualimetry was loaded into the knowledge base, first of all, the works of G.G. Azgaldova. The first PromT was developed to fulfill the procedures for compiling trees of the properties and conservation of calculations of weights of simple properties. Simple properties are the properties that can be measured analytically or expertly, and their weights are defined as a tanning coefficient (Y.N.K.).

To debugging Promt and confirming the possibility of using the STS, an experiment was conducted with the reproduction of an example from the book of prof. Azgaldova G.G. [2, 8] on the creation of a qualimetric methodology for determining the laureates of the National Franchising Award of the Golden Brand in the Golden Franchise nomination (as of the state of 2007). In the original example and complexity of work, it was estimated at about 9 people-days, excluding the costs of the head of the development of the methodology. Based on 19, the criteria was built, a tree of properties, represented in Table 1. The numbering of the properties of a warginal example was not applicable for the work of the bot, which is required to change the encoding.

Table **??**.

Next, a bot training was carried out in interactive mode, where errors were bothewed, and the relevant instructions were added to the PrOMT to prevent them. Thus, it was possible to train the bot to carry out the correct calculations of the lerteology of qualimetry. The final Prom amounted to 16,380 signs (including symbols., Smaching).

After receiving satisfactory training results, the bot was given to the downtime to reproduce an example from the book of prof. Azgaldova G.G. With interactive use with the bot, they revealed a certain discrepancy between the vine of the properties of the rules of composing such trees developed. Nevertheless, he was completely reproducing an example from the book and calculate the weight coefficients of the rules of qualimetry, which he did. The results of a comparison of the output parameter (Y.N.K.) and the values calculated by the bot is presented in Table 2

Table 1: Property tree from the example of efficiency analysis of franchise-nominated enterprises from the book by Azgaldov G.G.

| Level 0 | Level 1 | Level 2 | Level 3 | Level 4 | Level 5 |
|---|---|---|---|---|---|
| Franchise efficiency | 18. Results obtained - 1.1. | 16. Organizational work on franchises - 1.1.1. | 13. Generalized franchise characteristics - 1.1.1.1. | 12. Prevalence by number of franchises - 1.1.1.1.1. | 1. Number of franchisees with multiple franchises - 1.1.1.1.1.1. |
| | | | | | 2. Total number of franchisees - 1.1.1.1.1.2. |
| | | | | 3. Regional prevalence - 1.1.1.1.2. | |
| | | | 14. Franchise agreement success rate - 1.1.1.2. | 4. Number of terminated franchises - 1.1.1.2.1. | |
| | | | | 5. Percentage of terminated franchises - 1.1.1.2.2. | |
| | | 6. Total sales volume - 1.1.2. | | | |
| | 19. Generalized cost characteristics - 1.2. | 17. Franchising costs - 1.2.1. | 7. Franchisor's expenses on franchise advertising - 1.2.1.1. | | |
| | | | 15. Franchisor's expenses on franchise promotion - 1.2.1.2. | 8. Costs for franchisee support - 1.2.1.2.1. | |
| | | | | 9. Sunk costs from franchise termination - 1.2.1.2.2. | |
| | | | 10. Goodwill enhancement costs in franchise advertising - 1.2.1.3 | | |
| | | 11. Share of franchising costs in total advertising expenses - 1.2.2. | | | |

As can be seen from the table. 2, the results obtained by the bot are very close, for example, screaming. Some inconsistency is explained by the fact that in the example of the value of the group coefficients, they were averaged according to experts, while the bot appointed nimbly. It is noteworthy that the average data of experts in some cases were used to prostrate the rules of qualimetry. For example, in the decomposition of the parental property, "The success of the conclusion of the franchises" (Table 3), the experts gave directly opposite assessments, which may indicate questions for their competence. In such cases, the rules of the second round of assessment are required to coordinate the positions of experts, which was not made, which led to an error in the calculations of the example. In this delay, the calculation of the bot was more correct and relevant to the Rules.

Thus, in the first approximation, it was proved that the bot can correctly, a screw -mode, create trees of properties and conduct the necessary calculations of qualimetry corrective. This conclusion made it possible to proceed to assess the value of the territories. For this goal, a bot in another advanced version of the Gemini 2.0 Flash corporation Google was created. The created Prompt was transferred to the "qualimetryGM" bot and the materials on the theory and practice of qualimetry and the assessment of the territories. The bot was the task to build a tree of properties and make calculations of weight coefficients (Y.N.K.), provided that the three basic properties of value are equal to: economic, environmental and social. The interactive regime was turned off, and the bot should have been sagged to the parental properties to simple, measured, and manufacturing. To solve the problem of the bot, it was sufficient to decompose for 3 yarus (Table 4).

Conclusion

1. Although the STS has significant potential for transformationQualimetric analysis, a complete replacement of expert people at this stage is presented with malae.To reproduce the nuances of human judgment, implicit knowledge, common sense to and create complex, contextual-dependent situations. The most likely script. The future is a hybrid approach in which the STS is used as a powerful tool for complementing and expanding the capabilities of expert people. AIS canotomatize rulfinas, processed -large -lubricantsProvide initial results or offers (for example, quality criteria for weight). This allows experts to focus on more complex, strategic shoes of analysis, requiring a deep objective understanding and critical thinking.

2. Development of effective interactive Promov plays a key roleIn the direction of AI models to perform the tasks of qualimetric analysis with high flow and relevance. ITERIVE PROMES and Feedback cycles, possibly the sector of a person-expert, can help in improving work.

3. The integration of the STS into qualimetry raises important ethical andPractical questions. It is necessary to clarify the issues of accountability for errors, ensure the transparency of assessments based on AI, and take into account the risk of perpetuating or displacement of displacements from training data. Practical aspects include accessibility and the quality of training data, the necessary computing resources and training -users. The growing role of AI will require skills adaptation experts, including production development, testing models and solving complex ethical dilemmas. It is possible to maintain supervision and testing from a person in hybrid approaches to enforce accuracy, reliability and ethical use.

4. Accounting for multifaceted value along with the economic cost isThe key factor in the success and long -term sustainability of projects of complex development of territories. The scymetry.quantitative assessment of various aspects of value and their integration in the process of receiving decisions. The article shows and proves the fundamental possibility of II in the form of the STS for conducting research and measurements using the qualimetry. The following studies should be aimed at develop-ingpractical methods for assessing and integrating value into the practice of qualimetric measurements of value.

Table 2: Comparison of Qualimetric Analysis Results: Example vs. Bot Calculation

| Indicator | Tree Node | Example from Book | | | | |
|---|---|---|---|---|---|---|
| | | Expert Avg. | Norm. Gi | Calculated YNK | Code | Bot W |
| Franchisees with multiple franchises | 1 | 74 | 0.426 | 0.0637 | 1.1.1.1.1 | |
| Total franchisees | 2 | 100 | 0.574 | 0.0858 | 1.1.1.1.2 | |
| Regional distribution | 3 | 77 | 0.435 | 0.1153 | 1.1.1.1.2 | |
| Terminated franchises | 4 | 51 | 0.456 | 0.0720 | 1.1.1.2.1 | |
| Termination ratio | 5 | 61 | 0.544 | 0.0860 | 1.1.1.2.2 | |
| Total franchise sales | 6 | 47 | 0.324 | 0.2022 | 1.1.2. | |
| Franchisee support costs | 8 | 100 | 0.778 | 0.0972 | 1.2.1.2.1 | |
| Contract termination losses | 9 | 29 | 0.222 | 0.0278 | 1.2.1.2.2 | |
| Franchisor marketing expenses | 7 | 60 | 0.290 | 0.0937 | 1.2.1.1 | |
| Goodwill enhancement share | 10 | 67 | 0.324 | 0.1048 | 1.2.1.3 | |
| Franchising cost share in marketing | 11 | 16 | 0.138 | 0.0516 | 1.2.2. | |

*Note: YNK = Tiered Normalization Coefficient; Verification for Tier 5 = 1.0000*

Table 3: Franchise Success Rate

| Indicator | Tree Node | Non-Normalized Group Importance Coefficients (Experts) | | | | | | | Mean |
|---|---|---|---|---|---|---|---|---|---|
| | | Exp.1 | Exp.2 | Exp.3 | Exp.4 | Exp.5 | Exp.6 | Exp.7 | |
| Terminated franchises | 4 | 10 | 100 | 10 | 20 | 20 | 100 | 100 | 51 |
| Termination ratio | 5 | 100 | 10 | 100 | 100 | 100 | 10 | 10 | 61 |

Table 4: Economic Value Indicators for Comprehensive Territory Development

| Tier 0 | Tier 1 | Tier 2 | Tier 3 | WNC |
|---|---|---|---|---|
| Territory Value for CDP | Economic Value (100%) | Local Economy Diversity (100%) | Retail facilities availability | 0.0289 |
| | | | Service sector development | 0.0260 |
| | | | Food service representation | 0.0231 |
| | | | SME presence | 0.0202 |
| | | Business Infrastructure (90%) | Office spaces/coworking | 0.0327 |
| | | | Telecom quality | 0.0294 |
| | | | Business service accessibility | 0.0261 |
| | | Transport Connectivity (80%) | Public transport network | 0.0231 |
| | | | Pedestrian infrastructure | 0.0208 |
| | | | Cycling infrastructure | 0.0184 |
| | | | Road accessibility | 0.0161 |
| | | Job Creation Potential (70%) | Available business spaces | 0.0256 |
| | | | Entrepreneurial appeal | 0.0230 |
| | | | Labor market alignment | 0.0205 |

Designed in the article "Claramimetry" "You are an expert on qualimetry and build a tree of properties for solving a definite problem. It is necessary to build a tree of properties according to the laws of qualimetry of the peremns until the simple properties that can be measured. 1. Always ask the user at the beginning of the dialogue: "Remind you of the rules for compiling the properties according to the methodology of prof. Korostelev SP". Wait for the user's reaction! You will continue only after the user's reaction/no → if "no" go to paragraph 2, and if "yes", we display the text: "Rules for creating properties of properties" 10 key rules for compiling trees of properties according to the methodology: Determination of the main parental property (GRS) 1. GRS is an initial problem or a goal that is on tier 0. It always develops a weight of 100

8. The use of expert estimates of the purpose of weights and verification of the correctness of the tree involves experts with knowledge in the subject area. 9. Documentation of changes in the change in the tree of properties is recorded with the date and cause of the correlation. This ensures transparency and traceability of the process. 10. A check for completeness and consistency of the core should cover all aspects of the problem being studied and not maintain protesters between properties on different tiers: 10.1. The coverage of all key aspects of properties should include all the significant aspects of the problem under study, such as functionality, reliability, ease of use, cost and others. For example, for a CRM system, this can include not only its functionality, but the interface, speed of work, the cost of maintenance and other important characteristics. 10.2. The exclusion of contradictions by all levels of the tree must exclude the contradictions between the properties. For example, if one property implies high cost, and the other is low, lays coordination and clarifications. Contradictions can also arise between the properties of the worldly levels, therefore it is important to check their relationship and logical consistency. 10.3. The check for completeness should be complete, i.e.

Table 5: Environmental Value Indicators

| Tier 0 | Tier 1 | Tier 2 | Tier 3 | WNC |
|---|---|---|---|---|
| **Territory Value for CDP** | | | | |
| | Environmental Value (100%) | Environmental Quality (100%) | Air pollution level | 0.0289 |
| | | | Water quality | 0.0260 |
| | | | Soil contamination | 0.0231 |
| | | | Noise pollution | 0.0202 |
| | | Green Spaces (90%) | Green area per capita | 0.0327 |
| | | | Green zone coverage | 0.0294 |
| | | | Green space quality | 0.0261 |
| | | Biodiversity (80%) | Flora diversity | 0.0260 |
| | | | Fauna diversity | 0.0289 |
| | | | Protected species | 0.0231 |
| | | Energy Sustainability (70%) | Renewable energy share | 0.0256 |
| | | | Building energy efficiency | 0.0230 |
| | | | Resource use efficiency | 0.0205 |

Table 6: Social Value Indicators

| Tier 0 | Tier 1 | Tier 2 | Tier 3 | WNC |
|---|---|---|---|---|
| **Territory Value for CDP** | | | | |
| | Social Value (100%) | Social Infrastructure (100%) | Education facilities | 0.0265 |
| | | | Healthcare facilities | 0.0265 |
| | | | Culture/sports facilities | 0.0239 |
| | | | Social welfare facilities | 0.0212 |
| | | Safety (90%) | Crime rate | 0.0327 |
| | | | Perceived safety | 0.0294 |
| | | | Law enforcement efficiency | 0.0261 |
| | | Social Cohesion (80%) | Public spaces | 0.0291 |
| | | | Community activity | 0.0262 |
| | | | Citizen participation | 0.0233 |
| | | Living Comfort (70%) | Maintenance quality | 0.0144 |
| | | | Housing affordability | 0.0160 |
| | | | Housing quality | 0.0144 |
| | | | Utility infrastructure | 0.0128 |

Cover all the necessary properties for the solution of the task. This includes both the properties of the purpose of the object and its functional, operational and other characteristics. For example, for a tent, this can include protection against moisture, the strength of the material, the convenience of installation and other aspects. 10.4. Accounting for all aspects of the problem of the properties of the properties should take into account all the key aspects of the problem under study. This is both the properties of the purpose of the object and its functional, operational and other characteristics. For example, if an object is a CRM system, a tree is not only to use its functionality, but also the convenience of use, reliability, service cost and other important aspects. 10.5. The exclusion of the contradictions of all tiers of the tree must exclude the contradictions between the properties. For example, if one property implies high cost, and the other is low, we currently coordinates approval and clarification. Contradictions can also arise between the properties of a worldly tiers, therefore it is important to check their relationship and logical consistency. These rules provide a systematic approach to building trees and Ichispolization to solve multi -criteria problems. ”

Always turn on all points and subparagraphs of the rules, including 10.1, 10.2 and so on. 2. Now we need to decompose the complex integral property that is located to the tier 0 and does not have the encoding defined in the name of the problem. To clarify the context of the task before the decomposition: to clarify the context of the task and priority aspects before the decomposition of the integral property in order to recover the discrepancies in the substantiation of the weights. School 1: request the user's context of the task. Specify which aspect is a priority for the task: economic, environmental, social or other? This will help to correctly determine the key factors for decomposition. "Step 2: Offer decomposition options based on an updated context. Example:" If the economic aspect is a priority, the key factor will be economic attractiveness. Fix the selected approach. Example: “Selected approach: Economic attractiveness as a key factor. Fixing date: 2025-04-02.“ Checking data: Make sure that the context of the task and priority aspects are agreed by the Spain. Care that the selected approach is recorded and used for all subsequent steps. The format of the conclusion: an updated context of the problem. Approach to decomposition. Fixing of the approach. Example: enter the context of the task: “Evaluation of the territory for investment.“ Specify the priority aspect: “Economic.” Select the approach: “Economic attractiveness as a key factor.“ Complement the approach: “Selected approach is recorded. Date: 2025-04-02.“ Error messages: error 1: Error 1: Error 1: Error 1: Error 1: Error 1 "The context of the problem is not specified. Please specify the priority aspect." Error 2: "The selected approach is not fixed. Please fix the approach by continuation."

Recommendations: The selected context always take into account before the start of the decomposition. Fix the selected approach and use it for all subsequent steps. If the context of the problem changes, create a new decomposition based on updated dummies. Excert “how many quasi properties you need to decomle integration” and present the user for approval. And after the answer, the performance and assignment of the property with the property (for the first tier, for example, 1.1 and 1.2 for the separation of 2 quasi properties) and, after consistent with the user, weight - by the name of the value of 100

For each simple property, check the chain of normalized weights from it to the GRS. Reduce the standardized weights in the chain and get Y.N.K. Carry and correct errors with the standardized weightation: ensure the correct use of normalized weights when constructing the properties and calculation of Y.N.Ski execution: checking the standardized weights: if the normalized weight is equal 1.0000, this indicates an error, since this is possible for the GRS (tier 0). For all other tiers, the standardized weight should be less than 1.0000, and the summary of all quasi -free properties of the parental property should be equal to 1.0000. Marking: if a normalized weight equal to 1.0000, recount the formula: make sure that the amount of normalized weights All quasi -free properties of parental care is 1.0000. I.N.K. For all simple properties, using a chain of normalized scales from a simple property to the GRS. Crossing the amount of Y.N.K.: Make sure that the amount is Y.N.K. All simple properties are 1.0000 ± 0.01. Example use: Make sure that the final table with scales and I.N.K. Fixed cannot be changed. Before building a tree of properties, check that all normalized weights and Ya.N.K. are used to the rules. Documentation: if you need to make changes, create a new table and a tree of properties. Summarize all Y.N.K. And check if the amount is 1.0,000 ± 0.01. If the amount does not match, look for errors in the calculations or data. If the accuracy is unattainable, then bring all the calculations to the convenient table for verification in Excel - in all the calculated values, except for the code, put the comma instead of the point. 2. When working with a tree of properties, strictly observe the following rules: do not change the title of properties: all properties of

the properties should remain, as they are agreed and provided by the user. START DATA: Before making changes or adding new properties, check their suzeres with the consistent data. Cook changes: if the user makes changes, save them in a separate section or table with a indication of the date and table indicating the date and table. The reasons for the adjustment.

Checking for conflicts: if the discrepancy is detected between the current il -secured data, display a message indicating the conflict and requesting a confirmation of amendments. Documentation: All changes and updates record in the table with an indication: tier property of property of the property (

3. If accuracy is achieved, then bring the final results in the form of a readable table with citizens, columns and lines. In the first column, indicate the tier, in the second, the number in the third is the name of the property, in the third weight as a percentage, in the fourth -minute weight value. After each confirmation by the Customer: Save the table in the format: Property Code Violes of Venile (

Formatting: Borders of tables: | And-for dividers. Exposure: texts-on the left edge, number-by the right. Before making changes: Check new parameters with saved data. Example of conflict:* Current parameter: 1.1.1.2.1-"Number of franchise" (weight: 80

PrOMT to preserve the final table of table: the final table is stored in the memory as an unchanged object. All these tables (tiers, codes of properties, names, weights, normalized weights, Ya.N.K. Istatus) are fixed in the current state. Barrow of changes: Any attempts to change the table will deviate changes, the system must derive, the system must derive, the system must withdraw Message: "The final table is recorded and cannot be changed. If you requires the changes, create a new table." Access to the table: The user can request the output of the table at any time. The table is displayed in the format specified by the user (for example, with dividers | and

- for borders). Construction of integrity: before fixing the table, the system checks that the amount Y.N.K. It is $1.0000 \pm 0.01$. If the accuracy is not observed, the system displays the message: "The amount of Y.N.K. is not equal to 1.0000. Please check the data before fixing the table."

In addition to the final table, it is necessary to create another table only with simple. Columns: "Number in order, code, name, Y.N.K., Ya.N.K. Privates." . Before the output of the final table and the truncated table after each kodanado, put the point. Divide the numbers in Y.N.K. In all tables and Y.N.K. In percentage, replace instead of a point on a comma. You need to display these two tables to Excel. Approximately ask the user; "Do I need to convert the final table into the structure of themindmap? Yes/no." If the answer is yes, then: "1. ** Purpose: ** Transform the final table of properties into the MindMap structure with a complete decomposition of branches and verification of data correctness.

2. ** Fulfillment steps: **- ** Step 1: ** import the final table in CSV or Excel format.- ** Step 2: ** Build the hierarchical structure of the mindmap, starting from the root node (tier 0).- ** Step 3: ** for each tier:

Add properties in accordance with their codes and weights.- Make sure that all branches are laid out to simple properties.- ** Step 4: ** Check the correctness of the data:- The sum of normalized weights on each tier should be $1.0,000 \pm 0.01$.- the amount of Y.N.K. All simple properties should be $1.0,000 \pm 0.01$.- ** Step 5: ** Visualize the MindMap structure using tools (for example, minister, XMind, Coggle).3. ** Data verification: **- for each simple property, check the chain of normalized weights from it of the end of root node.- change normalized weights in the chain and get Y.N.K.- summarize everything Y.N.K. And make sure that the amount is $1.0,000 \pm 0.01$.4. ** The output format: **- Mindmap structure in text format with retreats and symbols '" '' '.- table with data verification (tier, property code, property name, weight (

Provide the correct use of scales from the final table in the construction of the MindMap structure.Steps of execution:Step 1: Before starting work, make sure that the final table with Libra and Y.N.K. K. K.The is refined and cannot be changed.

Step 2: When building mindmap, strictly use weights and Ya.N.K. From the final table. Irce them yourself. Shag 3: Check that all properties and their weights correspond to the data from the table. Shag 4: Make sure that the amount is Y.N.K. All simple properties are $1.0,000 \pm 0.01$. Shag 5: if a discrepancy is detected, stop the process and inform the error. Monitoring points:

The control point 1: The weight of all properties must strictly correspond to the final table. The control point 2: the amount of Y.N.K. All simple properties should be $1.0000 \pm 0.01$. The control

point 3: If the final table is changed, create a new mindmapon structure based on updated data.
Example use:

Enter the final table in the CSV or Excel format. Lower the process of building mindmap, strictly following the data from the table. Care that all weights and Ya.N.K. correspond to the table. The messages about the error:

Error 1: "The inconsistency of the scales was found. Please check the data in the final tightness."
Error 2: "The amount of Y.N.K. is not 1.0000. Please check the data with the transfixation of the table." Recommendations: Fixing data: Make sure that the final table is recorded and cannot be changed: Before constructing mindmap, check that all the weights are all weights and that all weight Y.N.K. Corresponding to Tabled. Documentation: If you need to make changes, create a new table and the MindMap renewal

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

# NeurIPS Paper Checklist

Reproduction: The work describes the proposed ML approach that combines Classic qualimetric methodology using the STS. It is mentioned that the conduct of research of the value of the territory was developed by the PrMPT, which is theOSTOBANNENTROURAL

Confirming the possibility of using the STS for automation of key procedures: building trees of properties and calculating values. It is described by experiment with a reproduction of an example from the book of prof. Azgaldova G.G. The compares of the qualimetric analysis from the example and calculations made by the bot (table2) is shown. It is indicated that for the development of Prombte, the environment of the integrator of the SGNSA DEPSEEK V3 SGNS integrator was used, and to assess the value of the territories - the Gemini 2.0 Flash12 STS. The bosom of the bot was loaded with work on qualimetry, first of all, by the works of G.G. Azgaldov, and materials on theory and practical -qualimetry and valuation of territories were added to evaluate the territories. The final Prompt amounted to 16,380 signs and presented the application.

Transparency: sources of data for training or the knowledge base of bots are disclosed - ecuphors on qualimetry, works of G.G. Azgaldova and materials on theory and practice and assessment of territories. The conclusions of the article explicitly indicate the need to deliberate the transparency of assessments based on AI, as an important ethical Ipractic issue. In the work presented, there are no potential conflicts of interests, financing of work from any sources except their own did not seem to be.

Ethics: The article emphasizes the importance of assessing multifaceted value (social, environmental, etc.) in addition to the traditional economic value for sustainable complex development of territories. The underestimation of these factors can lead to a casual, but negative projects from a long -term point of view. This is the awareness of the social and environmental impact of KRT projects. The integration of the STS in qualimetry raises important ethical and practical issues. The book includes:

The need to clarify the issues of accountability for errors.

The risk of perpetuating or enhancing displacements from training data. This is directly binded with the risk of discrimination, although the work does not detail how this can manifest itself in the assessment of the territories.

The need to take into account complex ethical dilemmas in the growing role of A

In conclusion, it is noted that the accounting of multifaceted value is a key factor of long -term stability of KRT projects, which corresponds to the ethical approach to Kgorod development.

Reliability: the reliability of the approach using the STS is checked by producing classic examples of qualimetric analysis. The results obtained by the bot are compared with the results from the example (table 2) for demonstrating intimacy. It is noted that in one case, the calculation of the bot was more correct and more consequent of the rule of fireimalimeters, which is average for the average person, ontaining an error due to violation of the rules. This may indicate the potential of JISSENTENTY OF Reliability by strict adherence to the methodology. However, the author is also guided by the restrictions of modern STS, such as the possibility of inaccuracies ("hallucinations") and difficulties with complex numerical reasoning. It is emphasized that the results of the STS require the verification of the output data of the model. It is important to maintain supervision and testing by a person in hybrid approaches to ensure accuracy, reliability and ethical use.

Human participation: Classical qualimetry requires the involvement of many experts, which is time -consuming. The ML approach with the STS is proposed for the automation of qualimetry and partial replacement of experts. However, the author directly argues that the complete replacement of expert people of the Nada stage seems unlikely. The most likely scenario is the future of a hybrid approach in which the STS is used to supplement and expand the abilities of expert people. AI automates routine tasks and provides the first results, allowing experts to focus on more complex adversary spaces. The development of effective interactive industrials and feedback cycles, possibly with the participation of a person-expert, can help improve the work of the STS.

The growing role of AI will require skills adaptation from expert people, including verification and the solution of ethical dilemmas. The importance of human supervision and verification of Viihbrid approaches to ensure accuracy, reliability and ethics is emphasized by the control. Unlike studies where experiments are conducted with people

353 Participants to study their behavior or reaction, this work describes the application of AI for the
354 performance of tasks that traditionally fill out experts.

355 Directly with experiments on people participating people, vistors are not considered.

