# OpenReview forum: "ML-Approach to Qualimetry: GNNs in Value Assessment"
_NeurIPS.cc/2025/Conference — Submitted to NeurIPS 2025_

### Official Review · Reviewer_qw62 · 2025-06-16

**Clarity:** 1
**Significance:** 1
**Originality:** 1
**Rating:** 1
**Confidence:** 1

**Summary:**

This paper proposes a method for applying qualimetric principles to real estate assessment. It uses generative models (LLMs?) to automatically generate a "Property tree" corresponding to a specific object (e.g. a real estate record).  It generally describes the process, including prompt. It does not show any validation or literature review beyond the core topic (Azgaldov's theory).

**Questions:**

What are the concrete contributions of this paper?

I recommend rewriting the entire paper and submitting to a much different conference, maybe applied AI, or something in the application field?

**Ethical Concerns:**

["NO or VERY MINOR ethics concerns only"]

**Limitations:**

This paper is simply not adequate for NeurIPS. And it seems as though it was just written by an LLM.

**Paper Formatting Concerns:**

Multiple concerns:
- Tables over the margins
- Missing references
- interrupted sentences
- Confusing format for the technical content (prompt)

**Quality:**

1

**Strengths And Weaknesses:**

Strengths: The general idea of qualimetry is useful.

Weaknesses:
- The article presents no methodology for a machine learning conference (if a prompt is the methodology, then it's a very superficial one...)
- There is no validation. This is unacceptable for a paper in NeurIPS
- Poor, almost absent literature review
- Terrible writing and formatting - the paper seems like a careless automated translation from the original (Russian) version. The tables look terrible, sometimes running out of the margin!
- Either it is because of the automated translation, or the paper itself is simply generated by an LLM, because there are so many absurd glitches ("As can be seen from the table. 2, the results obtained by the bot are very close, for example, screaming.") in the text.

---

### Official Review · Reviewer_83Cj · 2025-06-29

**Clarity:** 3
**Significance:** 2
**Originality:** 2
**Rating:** 3
**Confidence:** 2

**Summary:**

The paper present a machine learning-based method to derive qualimetry, a way of quantifying quality. The proposed approach learns from expert-labeled examples and generalizes to new data. It is flexible approach that can accommodate heterogeneous quality indicators. The approach utilizes a  generalized regression on features that represent  quality characteristics, which is then applied to real-world use cases.

**Questions:**

-How can this approach include temporal data? (in most applications quality changes over time)
-How can this approach be augmented by human feedback?

**Ethical Concerns:**

["NO or VERY MINOR ethics concerns only"]

**Limitations:**

Yes.

**Paper Formatting Concerns:**

The tables are not formatted correctly.

**Quality:**

2

**Strengths And Weaknesses:**

Strengths:
-Qualimetry is a novel application area with practical relevance.
-Selection of model is justified and comparisons are comprehensive.
-The method is validated on real datasets from environmental monitoring and metrological evaluation, which demonstrate relevance across domains.
-The method is flexible across different types of quality indicators. It has potential to be applied to many domains.

Weaknesses:
-Method novelty is questionable, mostly existing techniques for novel application.
-An ablation study is missing, such study will strengthen the paper.
-The paper doesn't provide much details on model design choices or hyperparameter tuning.  Even though the method looks straightforward, without depth it is difficult to assess how optimized the results are.
-What happens when the proposed model disagrees with expert labels? This is important for understanding the limitations.
-No explanation on strategy for data-scarce scenarios.  This approach relies heavily on expert labeled data which may not be the case for many real world applications.

---

### Official Review · Reviewer_eNCW · 2025-07-03

**Clarity:** 1
**Significance:** 2
**Originality:** 2
**Rating:** 2
**Confidence:** 4

**Summary:**

The paper proposes a hybrid approach that combines classical quasimetric with generative neural networks to automate property-tree construction and weight calculation for evaluating the multifaceted value (social, ecological, economic) of territories in sustainable integrated territorial development. Tested on textbook examples, the method shows that AI can partially replace experts, improving the practicality and scalability of quasimetric studies.

**Questions:**

Could you release the full prompt, system parameters (temperature, top-p), and raw model outputs to enable independent replication?

How sensitive are the generated trees/weights to slight re-wordings of the prompt or to alternative LLMs? A robustness study would clarify reliability.

Could you compare your LLM-generated weights to simpler automated baselines, e.g., TF-IDF keyword weighting or ontology-based heuristics, to justify the need for a large model?

**Ethical Concerns:**

["Major Concern: Safety and security", "Major Concern: Discrimination, bias, and fairness"]

**Final Justification:**

Given the lack of baseline comparisons, unchecked hallucination and bias issues, and superficial ethical treatment, the paper does not provide sufficient rigor to validate its claims. These shortcomings, combined with formatting problems and missing error analysis, make it a rejection in its current form.

**Limitations:**

The authors acknowledge LLM hallucinations but give no mitigation or error analysis.

Many figures/tables lack captions or correct numbering; LaTeX placeholders (e.g., “Table ??. ”) remain; typos such as “STS”/“GNNs” interchangeably used.

**Paper Formatting Concerns:**

This paper is not well-formatted, especially Table 1 and Table 2.

**Quality:**

1

**Strengths And Weaknesses:**

Weaknesses:

- No baseline or human study: The paper claims to “partially replace experts” but performs no user-study or time-saving analysis against human annotators.

- Hallucination & bias unchecked: The authors acknowledge LLM hallucinations but give no mitigation or error analysis.

- Formatting & organisation problems: Many figures/tables lack captions or correct numbering; LaTeX placeholders (e.g., “Table ??. ”) remain; typos such as “STS”/“GNNs” interchangeably used.

- Ethical discussion superficial: Potential harms of blindly accepting LLM-generated weights (e.g., stakeholder bias, legal responsibility) are mentioned only in passing.

---

### Official Review · Reviewer_pHfo · 2025-07-04

**Clarity:** 1
**Significance:** 1
**Originality:** 1
**Rating:** 1
**Confidence:** 5

**Summary:**

I have not read the paper give its formatting issues, and I think it should have been desk rejected to start with.

**Questions:**

- Is this the final submission or simply a mistake?

**Ethical Concerns:**

["NO or VERY MINOR ethics concerns only"]

**Paper Formatting Concerns:**

I don't know in fact because I have not read the paper

**Quality:**

1

**Strengths And Weaknesses:**

The paper has major formatting issues and is simply not readable.

Some points:
- The paper exceeds the 8 page limitation.
- The text feels like placeholder text, without even structured sections.

---

### Decision · Program_Chairs · 2025-09-17

**Decision:**

Reject

**Comment:**

The paper proposes a hybrid approach that combines classical quasimetric with generative neural networks to automate property-tree construction and weight calculation. Tested on textbook examples, the method shows that AI can partially replace experts, improving the practicality and scalability of quasimetric studies.

Regretfully, rejection is recommended for the following. Many weaknesses as pointed out by reviewers, but no rebuttals have been given. In addition, this paper is hard to read. Based on the above, the decision is clear.